# Multifunctional Plant Virus Nanoparticles for Targeting Breast Cancer Tumors

**DOI:** 10.3390/vaccines10091431

**Published:** 2022-08-30

**Authors:** Mehdi Shahgolzari, Hassan Dianat-Moghadam, Afagh Yavari, Steven N. Fiering, Kathleen Hefferon

**Affiliations:** 1Dental Research Center, Hamadan University of Medical Sciences, Hamadan 6517838636, Iran; 2Department of Genetics and Molecular Biology, School of Medicine, Isfahan University of Medical Sciences, Isfahan 8174673461, Iran; 3Department of Biology, Payame Noor University, Tehran 193954697, Iran; 4Department of Microbiology and Immunology, Dartmouth Geisel School of Medicine, Hanover, NH 03756, USA; 5Norris Cotton Cancer Center, Dartmouth Geisel School of Medicine and Dartmouth-Hitchcock Medical Center, Lebanon, NH 03756, USA; 6Virology Laboratory, Department of Cell & Systems Biology, University of Toronto, Toronto, ON M5S 3B2, Canada

**Keywords:** plant viruses, in situ vaccination, immunomodulatory agent, passive targeting, active targeting

## Abstract

Breast cancer treatment using plant-virus-based nanoparticles (PVNPs) has achieved considerable success in preclinical studies. PVNP-based breast cancer therapies include non-targeted and targeted nanoplatforms for delivery of anticancer therapeutic chemo and immune agents and cancer vaccines for activation of local and systemic antitumor immunity. Interestingly, PVNP platforms combined with other tumor immunotherapeutic options and other modalities of oncotherapy can improve tumor efficacy treatment. These applications can be achieved by encapsulation of a wide range of active ingredients and conjugating ligands for targeting immune and tumor cells. This review presents the current breast cancer treatments based on PVNP platforms.

## 1. Introduction

Breast cancer (BC) is the most common form of cancer in women that is associated with abnormal tissue growth within the breasts and has the potential to metastasize to other areas of the body such as bone marrow and lungs [1]. BC could be classified as hormone-positive (estrogen receptor, ER+, and progesterone receptor, PR+), human epidermal receptor-2-positive (HER2+), and triple-negative BC (TNBC) (ER−, PR−, HER2−) [2,3]. Various modalities such as surgery (mastectomy and lumpectomy), chemotherapy, radiotherapy, hormonal therapy, and immunotherapy have been used for the treatment of BC [2]. A significant shortcoming associated with these therapies is the lack of complete efficiency due to the heterogeneity of tumor cell biology, the complexities of the tumor microenvironment (TME), and several host factors [2,4]. Thus, novel therapeutic strategies for the effective treatment of BC are urgently required.

A new promising strategy for the effective treatment of BC is the coupling of therapy modalities with nanotechnology [5]. Nanotechnology can provide effective therapies through the design of multifunctional nanoparticles (NPs) with variations in size, shape, composition, or surface chemistry and their potential for targeting tumor cells in a specific manner [4]. One type of nanotechnology discussed in BC therapy includes NPs which can be used as the nanocarriers for the delivery of a variety of antitumor payloads and bioactive compounds to enhance their solubility, circulatory half-life, and biodistribution while reducing their unwanted side effects [6]. Furthermore, NPs could be functionalized with targeting ligands that specifically bind to receptors on the surfaces of tumor cells and precisely target BC cells [7]. In this regard, various NPs with synthetic and natural origins have been developed, among them natural biological building-block-based NPs such as protein cages and viruses are a promising class [8].

As a type of nanoparticle, plant-virus-based nanoparticles (PVNPs) can become the leading tumor-therapy approach. PVNPs are two subsets, virus-like particles (VLPs) and virion nanoparticles (VNPs). VNPs are self-assembled nucleoprotein structures based upon identical coat proteins (CPs) and nucleic acid and can form various morphologies, the most common being icosahedral, road-shape, and filamentous [9,10].

Plant-virus-derived VLPs are proteins’ cage-like sphere structures lacking the viral genome [9,11]. Platform technologies based on PVNPs and VLPs are compelling due to [9,12,13]:Their inherent safety, their non-infectious nature in mammals, biocompatibility, and biodegradability;Their well-defined structural features, such as unique shapes and sizes, can be monodispersed for loading targeted molecular antitumor therapies onto their internal cavity and their interior and exterior surfaces by the assembly, infusion, or internal surface modification;Their ability to self-assemble;Their ability to be localized to the target site by chemical and genetic programmability;Their simple and inexpensive production;Their inherent immunogenicity enables them to act as nano-adjuvants and nanovaccines in cancer immunotherapy

In addition, morphological uniformity, water-solubility, and administration doses of up to 100 mg per kilogram body weight without signs of toxicity are unique advantages of PVNPs over other synthetic NPs [8,14]. This review discusses how PVNP and PVNP-based strategies could be a promising nanoplatform for BC treatment via therapeutic agent delivery, cancer immunotherapy, vaccines, and combination therapies.

## 2. Multifunctional PVNPs in Cancer Therapy

The use of PVPNs in tumor therapy research has been developed. They are widely used as therapeutic agent carriers by encapsulating and protecting them from degradation. The surface of PVNPs can be modified with the relevant ligands for targeting cancer cells (Figure 1A) [13]. These PVNPs can display the tumor-specific antigens as a vaccine and with exogenous adjuvant delivery can improve the efficiency of antitumor responses [12,13,15]. Furthermore, some PVNPs can act as adjuvants and with their monotherapy can modulate immunosuppressive TMEs via ligands for pattern recognition receptors (PRRs) of immune cells and induce the production of cytokines and chemokines (see Section 3.3) [12,16].

Nanoengineering of PVNPs via noncovalent and covalent strategies has been developed specifically for loading and retaining therapeutic cargos and target molecules on PVNP surfaces. Noncovalent methods for packaging various cargoes rely on disassembly and reassembly to form uniform hollow containers (self-assembly method), electrostatic charges and hydrophobicity/hydrophilicity interactions (incubation method), and pores of “open” and “closed” states under certain environmental conditions (infusion method). Covalent-based cargo-loading methods take advantage of functional addressable groups (i.e., conjugation methods) and express simultaneously or separately by heterogeneous expression systems followed by self-assembly in vivo or in vitro (i.e., genetic methods) (reviewed in [8,9,12,13].

## 3. PVNP-Based BC Tumor Therapies

Similar to other solid tumors, the breast TME is composed of interactions between cellular and non-cellular compartments [17,18,19]. This TME acts as a barrier against current tumor treatment approaches and plays a critical role in BC development and progression as well as in determining the response to therapy [17]. PVNPs can be designed in a variety of ways to target these tumor barriers more importantly via the enhancing distribution of therapeutic agents into tumorous tissue and the fine-tuning of immunological responses [19]. The tumor targeting aspect of PVNPs relates to the inherent properties of PVNPs including [9,13]:Nanoparticulate features such as composition, size, and surface properties;Loading and targetability via nano-engineering;Inherent immune stimulatory ability.

Moreover, the antitumor activity of PVNPs could be boosted by the inherent tumor properties including [20]:The nature of tumor-associated vasculature in comparison to normal vasculature;Overexpression of tumor-cell-based biomarkers in comparison to normal cells;

Overall, the current trend of PVNPs in BC treatment include: (1) the use of PVNPs as nanovehicles for non-targeted delivery (passive targeting) that rely on extravasation and increased concentration of PVNPs in tumors, (2) targeted delivery (active targeting), coupling PVNPs to targeting molecules overexpressed on tumor cells to efficiently infiltrate tumors and enter malignant cells, (3) tumor-targeted immunotherapies, and (4) combinational therapies.

### 3.1. PVNP-Based Non-Targeted Delivery for BC Tumor

Non-targeted delivery is reliant upon the composition, size (ranging from 50 to 200 nm), and surface properties of PVNPs, which all increase the permeability of the endothelial blood microvasculature of tumors, (i.e., enhanced permeation and retention (EPR) effect) [21]. For implementation, the particulate nature of PVNPs enables antitumor therapeutic payloads to be held within the interior cavity or on the exterior surface and become passively recruited to the tumors by the EPR effect (Figure 1C) [9,16]. In addition, PVNPs have blood and tissue compatibility, are stable under physiological conditions, and are less prone to interact with serum proteins compared to synthetic nanoparticles [22,23]. PVPNs are protein aqueous, imparting better cell penetration and endolysosomal escape than some synthetic NPs [22]. These properties of tumorous vascular and PVNPs improve passive targeted delivery and lead to the enhancement of the therapeutic index, the circulation time, the efficiency of delivery, and enhanced payload accumulation within solid tumors [24].

However, rapid blood clearance by reticuloendothelial system (RES) cells and protein corona effects can reduce the efficacy of non-targeted delivery of PVNPs before reaching the target site [25,26,27]. In most cases, the decoration of PVNPs with serum albumin or polyethylene glycol (PEG) results in a ‘camouflage’ effect, preventing their antibody recognition, thus enhancing their pharmacokinetics [28,29]. Furthermore, compared to their spherical counterparts, the shape of PVNPs is analogous to elongated architectures of nanomaterials, enhancing tumor homing and retention properties [30,31,32]. These properties can be due to increased margination toward the vessel wall, which will present ligands more effectively to target cells, and the flat vessel wall and are more likely to resist immune detection and macrophage uptake and contribute to longer circulation times, and beneficial flow properties. In addition, the short, cross-sectional dimension of nanorods determines membrane transfer efficiency [14,33]. Filamentous (e.g., potato virus X) or spherical (e.g., cowpea mosaic virus) shape mirrors the phenomenon that the human tumor xenografts exhibit higher uptake of PEGylated filamentous PVX compared to spherical CPMV [30].

Some antitumor therapeutic agents such as small molecule drugs, nucleic acid, and peptide/protein polymers loaded onto PVNPs have been used for delivery to breast tumors as summarized below.

#### 3.1.1. Small Molecule Drug Delivery

The limitations of cancer treatment with small molecule drugs include poor bioavailability, drug resistance, recurrence, rapid drug clearance, non-targeted administration, high toxicities, and other side effects [34]. Therefore, various PVNPs have been developed and used for reducing these limitations. For example, loading mitoxantrone (MTO), a topoisomerase II inhibitor; phenanthriplatin, a cationic monofunctional DNA-binding platinum (II) complex; and gemcitabine, one of the nucleoside analog drugs, into tobacco mosaic virus (TMV) increased their accumulation within the tumor tissue and induced sufficient cytotoxicity toward MDA-MB-231, TNBC, and Michigan Cancer Foundation-7 (MCF7) cell lines [35,36]. In vivo MTO-TMV delivery in a mouse model of MDA-MB-231 xenografts reduced tumor growth, showing superior efficacy over free MTO [35]. Phenanthriplatin-loaded TMV in the MDA-MB-231-bearing mouse model has shown that the amount of drug within the tumor, when delivered by TMV, was increased ∼10-fold compared to the amount of drug administered systemically [37]. Furthermore, TMV-based spherical nanoparticles (SNPs), TMV, and Johnson grass chlorotic stripe mosaic virus (JgCSMV), loaded with doxorubicin showed a sustained drug release profile in BC cell lines (MDA-MB-231 and MCF-7) and breast tumor models [38,39]. Intraperitoneal (IP) injection of an MCF-7 tumor-bearing athymic mouse model with FA-JgCSMV-Dox tumor volumes were smaller when mice were treated with FA-JgCSMV-Dox compared to the controls [39]. Loading the prodrug 6-maleimidocaproyl-hydrazone doxorubicin (DOX-EMCH) with Physalis mottle virus (PhMV)-based VLPs and coating with PEG resulted in significantly great antitumor efficacy in vivo [40]. Overall, PVNPs present a programmable nano-scaffold-based platform for developing chemotherapeutics for BC models.

#### 3.1.2. Nucleic Acid Delivery

Delivery of antitumor therapeutic nucleic acid polymers to BC can be limited due to extracellular and intracellular barriers such as poor cell uptake, instability in circulation in the presence of nucleases, and non-efficient delivery to their target site [23,41]. For example, ‘naked’ small interfering RNA (siRNA) is not stable in plasma, not targeted, and their negative charge impairs efficient cell uptake [42]. On the other hand, the use of nonviral nanocarriers composed of lipids or polymers often does not match the efficiency achieved using biological systems [23]. Therefore, the efficient delivery of this type of agent demands highly innovative systems [43]. The inherent nature of PVNPs to carry and deliver their genome into the host cells make them suitable as nanocarriers and an alternative or complementary approach for delivering therapeutic nucleic acids into tumor cells [43,44]. Importantly, they are preferentially taken up by innate immune cells such as macrophages (mainly M2 macrophages or tumor associates macrophages, TAM) and dendritic cells (DCs) in the TME [22] or can act as a toll-like receptor (TLR) agonist [45]. All these properties allow the PVNPs to be utilized to deliver therapeutic nucleic acids to innate immune cells in the treatment of cancer [8].

Recently, various PVNPs were used for encapsulation and delivery of nucleic acid polymers such as heterologous RNA, siRNAs, mRNA, and oligodeoxynucleotides (CpG-ODNs) into mammalian cells [22,23,42,46,47,48]. Akt 1 is a kinase involved in the processes of tumor cell proliferation and migration, and two examples, brome mosaic virus (BMV) and cowpea chlorotic mottle virus (CCMV) loaded with the antitumor siRNA Akt1 (siAkt1), have shown to be capable of being internalized by the tumor cells and thus, able to deliver the siAkt1 cargo into breast tumor cells [49]. CCMV formulated with siRNAs targeting FOXA1 (as a transcription factor of the Forkhead box (FOX) protein family) would allow gene silencing using the BC cell line MCF-7 [42]. In conclusion, besides enhancing the cellular uptake efficiency, the application of the PVNP-CP/VLP delivery system protected therapeutic nucleic acid from digestion while presenting better biocompatibility compared with the free formats [50].

#### 3.1.3. Peptide/Protein Delivery

Amino acid polymer-based therapeutics are easy to synthesize, have high target specificity, and have low toxicity. When compared to conventional cancer treatments, amino acid polymer-based treatments are a promising novel approach to the treatment of cancer [51]. However, their application faced several drawbacks such as low stability, short half-life, and difficulties with encapsulation in vitro [8,51]. PVNPs can apply a framework for improving these problems via chemical or genetic fusion and self-assembly of the viral capsid protein around these therapeutic agents [51]. For example, Herceptin (trastuzumab), a humanized monoclonal antibody (mAb) that targets human epidermal growth factor receptor 2 (HER2)-positive suffers from a short half-life, structural heterogeneity, instability, and solubility limitations [44]. Herceptin or HER2 epitopes (CH401 epitope) conjugated with PVNPs such as PVX, Physalis mottle virus (PhMV), and CPMV have shown the promising targeting potentials of these platforms [52,53,54,55].

In addition, PVNPs can act as scaffolds for the delivery of proteinous drugs. For example, PVX, CPMV-Herceptin (trastuzumab) compared to free Herceptin significantly increases the rate of apoptosis in HER2 positive cell lines [52,56]. The conjugation of VEGFR-1 ligand and fluorescent PEGylated peptide on CPMV NPs can target VEGFR-1 on endothelial cell lines and VEGFR1-expressing tumor xenografts in vivo [57]. TRAIL, part of the tumor necrosis factor (TNF) superfamily is expressed as a homotrimeric type II transmembrane protein or under proteolytic cleavage converted into a soluble form [58]. TRAIL binds to the death receptors (DR4 and DR5) resulting in receptor trimerization and recruitment of FAS-associated protein with the death domain (FADD) to activate the extrinsic or caspase-dependent apoptosis in cancer cells (but not healthy cells) [59]. A short half-life, instability of the monomeric form of the TRAIL protein, and rapid renal clearance of the off-targeted TRAIL are the most significant obstacles to effectively triggering apoptosis in cancer cells [60]. Meanwhile, conjugating TRAIL protein to PVX could mimic the native TRAIL function, activate caspase-mediated apoptosis more efficiently compared to soluble TRAIL, and delay tumor growth in human TNBC xenografts [58].

### 3.2. PVNP-Based Targeted Delivery for BC Tumor

PVNPs have intrinsic ligands for binding to immune cells; however, they have no ligands on their surfaces for targeting and binding to cancer cells for payload delivery [8]. Thus, targeted ligands must be incorporated into the VLP or capsid scaffold of PVNPs for binding to specific receptors overexpressed on cancerous cells to maximize payload efficacy (Figure 2A). BC cells overexpress particular biomarkers such as the tyrosine kinase epidermal growth factor receptor (EGFR) (also known as ErbB-1 or HER-1), folate receptors (FR), and human HER2 [7]. Ligands of these receptors can conjugate to the surface of PVNP via nano-engineering to specifically target cancer cells. The mechanisms of ligand-based targeted PVNP for efficient treatment can include receptor-mediated endocytosis, receptor-mediated block, and receptor-mediated activation.

EGFR as the receptor tyrosine kinase (RTK)-based transmembrane receptor is overexpressed on the surface of BC cells (100 times more than normal cells) and also other solid tumors [7]. EGFR becomes activated with ligands of the EGF family and is internalized mostly via the clathrin-mediated pathway for triggering uncontrolled proliferation of cancer cells [61]. Therefore, EGFR is an ideal cancer biomarker for designing EGFR-based targeted therapies [62]. EGF-based mimetic ligands that do not activate EGFR-mediated signaling but are conjugated to PVNPs carrying toxic payloads is particularly promising [61]. Recently, GE11, a small peptide with 12 amino acids with imaging moieties (A647-PVX-GE11), was displayed on the surface of PVX and indicated suitable targeting and imaging EGFR+ MDA-MB-231 of ductal breast carcinoma (BT-474) cells [61,63].

Folate receptors (FRs) are membrane-bound surface proteins with a high expression on the surface of BC cells that have high affinity to folates and folate-conjugated therapeutics [7,64]. Recently, FR-targeted drug-loaded NPs have received great interest due to their enhanced tumor and tumor cell uptake [7]. Strategies for targeted drug delivery to an FR-overexpressing tumor cell with a PVNP include:Delivery of FA-targeted PVNP loaded with anticancer therapeutic agents [39,65,66,67];FR-based drug agonists (e.g., MTX) conjugated to PVNPs;Dual targeting by presenting FA and conjugation of drug agonist (e.g., MTX) on surface PVNPs [68].

For example, conjugation of FA onto the capsids of hibiscus chlorotic ringspot virus (HCRSV) and CPMV has the potential for the targeted delivery of cancer chemotherapeutics [65,66]. Loaded doxorubicin in the inner cavity of CCMV nano-vehicles and conjugated with FA show selective delivery and cytotoxicity in FR positive MCF7cells in comparison to FR negative HepG2/HEK cell lines [67]. Furthermore, in comparison to free Dox, the FA- JgCSMV-Dox significantly reduced the tumor growth and cardiotoxicity of athymic mice bearing human BC xenografts [39]. It was also shown that more efficient than free paclitaxel (PTX), the pepper mild mottle virus (PMMoV)-based FA-targeted rod-shaped NPs loaded with PTX, FA@ PMMoV@PTX induced cytotoxicity against MCF-7 cells [69].

HER2, a membrane tyrosine kinase, is overexpressed in BCs to stimulate tumor growth by induction of anti-apoptosis signals [70,71]. HER2 is an important biomarker for targeting approximately 20–30% of human BC [72]. The HER2 ligands include trastuzumab which is very effective in treating HER2 + BC, with a 50–80% response rate [70]. However, they do not respond to trastuzumab therapy due to resistance after one year of treatment [53,70]. Recently, Herclon, an mAb against inside cell domain of HER2 receptor loaded by Sesbania mosaic virus (SeMV)-based VLPs fused with the B domain of Staphylococcus aureus protein A (SpA) effectively and could internalize into mammalian cells much more effectively than treatment with the antibodies alone [73]. The conjugation of trastuzumab to CPMV and the lysine mutant of TMV was recently shown to target and kill HER2-positive cells [56].

Moreover, a variety of BC cell receptors including estrogen receptors, CD44, transferrin-receptor, avb3 Integrin, biotin receptor, TRAIL receptor, and luteinizing hormone releasing hormone (LHRH) or gonadotropin-releasing hormone (GnRH) were developed for delivering ligand-tagged therapeutic NPs [7]. As mentioned above, using PVNPs as carriers can be promising as an alternative approach to increase TRAIL delivery (Figure 2B). Recently, it has been demonstrated that PVX-based targeted TRAIL activates caspase-mediated apoptosis more efficiently compared to soluble TRAIL in an athymic nude mouse model bearing human TNBC xenografts [58].

Neuropilins (NRP-1, 2) as the multifunctional non-tyrosine kinase receptors are expressed in tumor cells supporting angiogenesis and tumor growth [74]. In addition, overexpressed NRP-1 promotes cell invasion and lymph node metastasis [74,75,76]. Recently, it was demonstrated that functionalized TMV harboring a membrane-targeting peptide that targets the transmembrane domain of the Nrp1 receptor can bind to cultured MDA-MB2 cancer cells and inhibit NRP1-dependent angiogenesis in vitro [77].

Tumor-homing peptides (THPs) are safe, noncytotoxic to normal cells, and nonantigenic oligopeptides with a targetability to tumor cells for diagnostic and therapeutic applications [78]. Recently, the genetic fusion of the tumor-homing peptide F3 onto CCMV loaded with dye IR780 iodide, F3-CCMV-IR780 NPs, exhibited excellent in vitro targeted delivery, cellular uptake, and photothermal toxicity on MCF-7 tumor cells under the illumination of a near-infrared laser [79].

Finally, the low penetration of free therapeutic agents into solid tumors with dense extracellular matrix (ECM) could be augmented by targeting ECM-associated markers. Integrins are a large group of heterodimeric transmembrane glycoproteins that mediate the attachment of tumor cells to extracellular ECM molecules. Out of 24 reported integrin receptors, 8 recognize the RGD (arginine glycine-aspartic acid) sequence [80]. These eight receptors are responsible for various functions, including cell adhesion, cell survival, binding of blood platelets, cell migration, and angiogenesis [81]. For example, the genetic and chemical engineering of human adenovirus type 2 (HAdV-2)-derived RGD bearing CPMV particles interact strongly with both HeLa and HT-29 cells and internalize into them in vitro [82].

### 3.3. PVNP-Based Targeted Immunotherapies

PVNP-based targeted immunotherapies arise from their inherent immunogenicity and nanocarrier properties [12]. In this regard, PVNPs, VLPs, and spherical particles (SPs) can act as immunostimulatory agents, adjuvants, or vaccine platforms to activate the innate immune response [83,84]. PVNPs’ immunostimulatory properties can be dependent on structural properties (capsid protein and genome) that are non-self for the mammalian immune system and their organized regular spatial structures [83]. In addition, PVNPs present the pathogen-associated molecular patterns (PAMPs) to bind and active TLRs on the surface (TLRs 1, 2, 4, 5, and 6) or in the endosome (TLRs 3, 7, 8, and 9) of immune cells [45,85]. Furthermore, PVNPs also can improve the efficacy of other PAMPs with encapsulation and their targeted delivery to immune cells [10]. Therefore, PVNPs-based targeted immunotherapies are advantageous for they are multifunctional and multivalent platforms allowing large-scale manufacturing [72]. Moreover, PVNPs can act as vaccines without requiring additional adjuvants by displaying cancer antigens on their surface or encapsulating genomic material encoding the cancer-specific antigens [86].

PVNP-based monotherapy is performed through systemic administration of a single PVNP without any modification for induction of both innate and adaptive immune responses [12]. However, upon systemic delivery, PVNPs can reach the target area of tumor sites and be intercepted by the liver or spleen-resident phagocyte cells and then sequestered in these cells. This limitation could be alleviated through in situ vaccination (ISV) or direct intratumoral injection of PVNP into an identified tumor or metastatic site. In this regard, tumor APCs expressing TLRs recognize PVNPs to induce the production of cytokines and chemokines (e.g., IL-1β, IL-6, IL-12p40, MIP1-α, and TNF-α and type I interferon (IFN)) within cold TME and eventually reprogram the suppressed TME to an immune-activated state by converting TME-based TANs and TAMs from a protumor to an antitumor phenotype toward tumor eradication [87] (Figure 3A). In comparison with other plant viruses such as cowpea severe mosaic virus (CPSMV) and tobacco ring spot virus (TRSV), CPMV-ISV significantly activates TLR2 and 4, and also, induces the production of intratumoral IFNβ even four days after the second of two weekly treatments, confirming that CPMV can induce a memory response in a mouse model of dermal melanoma [88].

The antitumor immune stimulation of PVNPs could be induced via its packaged RNA, the multivalent nature of PVNP/VLP, or both that act as PAMP for the host immune system [89,90]. For example, ISV with CPMV and empty CPMV (eCPMV), a CMVA-free version of CPMV, have demonstrated efficacy in mouse models of melanoma, BC, ovarian cancer, and colon cancer [85,87,91]. The encapsulation of single strand RNA (ssRNA) or siRNA into PVNPs such as CCMV could be achieved through CP and CP–RNA interactions and using well-established, pH- and salt-controlled, dis- and assembly methods. Upon acidification, the fully ordered VLPs are formed to package CP/RNA complexes with suitable stability [92]. Addressing the BC, the preclinical data indicated the potent efficacy of PVNPs-ISV in the context of breast tumors in several mouse models. For example, eCPMV and CPMV therapies reverse the immunosuppressive TME in the 4T1 BALB/c BC model and promote the antigen-presenting ability of innate immune cells; therefore, restarting the cancer immunity cycle causes potent tumor regression [87,93]. Most recently, we investigated the therapeutic effects of alfalfa mosaic virus (AMV)-based ISV on the 4T1 BC model, and results have shown that therapy increased costimulatory molecules, inflammatory cytokines, and immune effector cell infiltration while downregulating the immune-suppressive molecules [94].

PVNP-based cancer vaccines include tumor antigens that are linked to the external surface of PVNPs or integration of related genomic material encoding the antigen encapsulated to the interior cavity of PVNP/VLP [86] (Figure 3C). Vaccination leads to the uptake of PVNPs by antigen-presenting cells (APCs) or specific receptors on B cells even in the absence of extra adjuvants [70]. Frequent administration of free trastuzumab with an in vivo short half-life as a type of passive immunotherapy does not protect patients from the development of tumor metastasis or recurrence. Meanwhile, PVNP-based vaccination is a promising candidate that can improve the limitations of passive immunotherapies and overcome the raised challenges by reducing self-tolerance and the number of administrations, while simultaneously enhancing the antitumor immune response and long-lasting immune memory [53,54]. It was demonstrated that HER2-derived B-cell epitopes (e.g., the CH401 peptide) conjugate to the icosahedral CPMV, and filamentous PVX, CCMV, and SeMV can break immune tolerance and induce the generation of antitumor antibodies that recognize HER2 on cancer cells [54,70,72]. Given the simplicity of design and manufacturing, such therapeutic vaccines based on the biocompatible CPMV platform technology could offer cost-effective and potent alternatives to current adjuvant therapies [72].

This type of bio carrier has a natural tropism for APCs through identification of key host proteins for a viral infection, such as site-1 protease (S1P), and thus, vaccination of PVNPs leads to activation of APCs, intracellular processing of loaded/decorated antigens, and the activation of CD4+ and CD8+ T cells. Activated DCs cells subsequently activate other immune cells with antitumor activity (e.g., T cells, NK cells, macrophages, and neutrophils). T cells can stimulate B cells to produce antibodies against tumor antigens (Figure 3). The efficacy of PVNP-based immunotherapies by CPMV, PVX, AMV, and TMV was demonstrated in several tumor types, including melanoma, ovarian, and BC [87]. In this regard, eCPMV particles can induce both DCs and macrophages to prime the CD4+ T cells in the mice model and also induce the CD11b+ Ly6G+ activated neutrophil subset expressing MHC class-II, indicating potential antigen presentation and T cell priming capability [87]. For example, ISV with CPMV PNVPs has shown that CPMV vaccination resulted in a reduced tumor burden and median survival of 81 days in a mouse model of human carcinoma, while other plant virus candidates (PhMV-VLP, SeMV, CCMV) and viral particles of non-plant origin (bacteriophage Qβ-VLPs and HBVc) examined here do not match the potency of CPMV [95]. Cucumber mosaic virus (CuMV)-VLP-expressing a hepatitis C virus (HCV)-derived epitope is another nano-sized particle that has the potential to trigger the generation of neutralizing antibodies, CD4+ T_H_ cells, and CD8+ T cells [96]. Incorporation of tetanus toxoid epitope TT830–843 into CuMV-VLPs boosted their immunogenicity [97], and the formulation of this platform with an adjuvant such as microcrystalline tyrosine (MCT) has shown to present a longer exposure time for the immune system and thus enhance the generation and filtration of antitumor CD4/CD8 T cell response against melanoma in the murine model [98].

Addressing adjuvant potency, PapMV NPs have shown to be a TLR7 ligand, a receptor that triggers innate immunity, and also production of IFNs to promote antitumor T cells activity [99] (Figure 3B). In this regard, DNA-containing plant viruses such as CaMV may have a stronger adjuvant effect than RNA viruses such as BMMV, of which the former can effectively stimulate IgG1, IgG2a, and IgG2b isotypes [83]. Therefore, effective, safe and low-cost adjuvants are necessary for modern vaccinology; thus, the PVNPs could be promising candidates for contributing to the design of novel adjuvants or vaccines for cancer immunotherapy [83].

### 3.4. PVNP -Based Combinational Therapies for BC

Tumor heterogeneity and immunosuppressive TME limited the therapeutic efficacy of monotherapy; thus, combining the introduced tumor therapies can be a promising strategy [12]. Combination strategies generally integrate mechanisms of therapies with a synergistic effect to overcome the complexity and tumor heterogeneity [100]. Inherent immunotherapy properties of PVNPs and their ability for loading therapeutic agents, immunomodulatory, and immune checkpoint agents offer new promise for combination immunotherapy, chemo−immunotherapy, and radiation immunotherapy now that their efficacy was demonstrated in mouse models of melanoma, breast cancer, ovarian cancer, and colon cancer [1,101,102,103,104,105].

A powerful approach can be chemo-immune combination therapy because cytotoxic drugs can induce cell death following the hallmarks of immunogenic cell death (ICD); thus a combination of tumor PVNP-mediated immunotherapies can increase antitumor efficacy [1]. For example, combination monotherapy of CPMV and cyclophosphamide (CPA)-based ICD against 4T1 mouse tumors increases the secretion of several cytokines, activates APCs, increases the abundance of tumor-infiltrating T cells, and systematically reverses immunosuppressive TME [1]. The 4-1BB (CD137, TNFRSF9) ligand is a trimeric membrane protein that is expressed on the surface of macrophages, activated B cells, and DCs. Activation of 4-1BB using 4-1BB-agonistic mAbs was found to trigger CD8+ T and NK cell activation as well as induce tumor regression. However, its clinical application has been held back by off-tumor toxicities and could therefore benefit greatly from the addition of tumor-targeting functionality to restrict its effect on the tumor deposits [106]. Moreover, the immunosuppressive TME reduces NK number and activity. Meanwhile, combinatorial therapy of CPMV and anti-4-1BB mAbs, which can recruit and activate NK cells in TME, respectively, can provide potent and durable antitumor efficacy as confirmed in solid tumor models in vivo [107].

Another strategy is the combination of PVNP monotherapy with immune checkpoint therapy. Specifically, anti-PD-1 antibodies or agonistic OX40-specific antibodies remove immunosuppressive T cells and thus generate a synergistic antitumor effect in mouse tumor models [103]. In addition, BC cells expressing CD47 ligand bind to signal-regulatory protein α (SIRPα) on phagocytic cells and thus protect them from cell-mediated phagocytosis. Therefore, administration of anti-CD47 antibody sensitizes BC cells to cell-mediated phagocytosis [108]. For example, it was demonstrated that CPMV-based ISV and CD47-blocking antibodies have the synergistic potential to induce tumor cell death through macrophage activation in the 4T1 breast tumor model [93].

Finally, the combination of immunotherapy and photothermal therapy (PTT) or radiation therapy is a potent strategy to improve cancer therapy efficacy. Recently, treating mouse dermal melanoma with the loading immunomodulatory 1V209, a TLR 7 agonist, into TM and coating with photothermal biopolymer polydopamine (PDA) highlights the potential of PVNPs as a multifunctional nano-platform for combined PTT-immunotherapy [109]. Furthermore, the combination of radiation therapy and CPMV-based ISV enhances efficacy over RT alone, in a mouse model [102]. Studies show that the combination of the PVNP-based therapies may be a particularly powerful strategy because chemo, RT, and PTT therapies can provide tumor antigens via immunogenic cell death, therefore synergizing with PVNP-ISV-enhanced antitumor immunity. In conclusion, BC cell survival and progression are regulated through multiple pathways and need to focus on these pathways by combining different therapeutic agents (e.g., immunotherapy and targeted therapy), demonstrating synergic effects to obtain better results.

## 4. Conclusions

Nanoparticles are being used in cancer treatment, often with particular focus on the delivery of therapeutic agents. However, systemic administration has shown that only approximately 1% can accumulate in TME, even with a high EPR effect. Thus, to make an effective impact, new delivery systems and delivery strategies are required. In this regard, PVNPs function as building blocks with the capacity to deliver therapeutic agents or as an in situ vaccination strategy, and the combination of them has shown promising results. Multifunctional PVNPs which load, protect, and control the targeted release of their cargo and are also inherently immunomodulatory, can facilitate immunogenic cell death and thus modify the tumor microenvironment. Studies show that PVNPs-based therapeutic agents’ delivery and in situ vaccination have shown synergistic efficacy against BC tumors in preclinical trials. Furthermore, they can combine PVNP in situ vaccination with multiple treatment strategies collectively into a single platform. Therefore, combination therapy may pave the way for a novel in situ cancer vaccine to give BC patients the best possible outcomes in the future.

## Figures and Tables

**Figure 1 vaccines-10-01431-f001:**
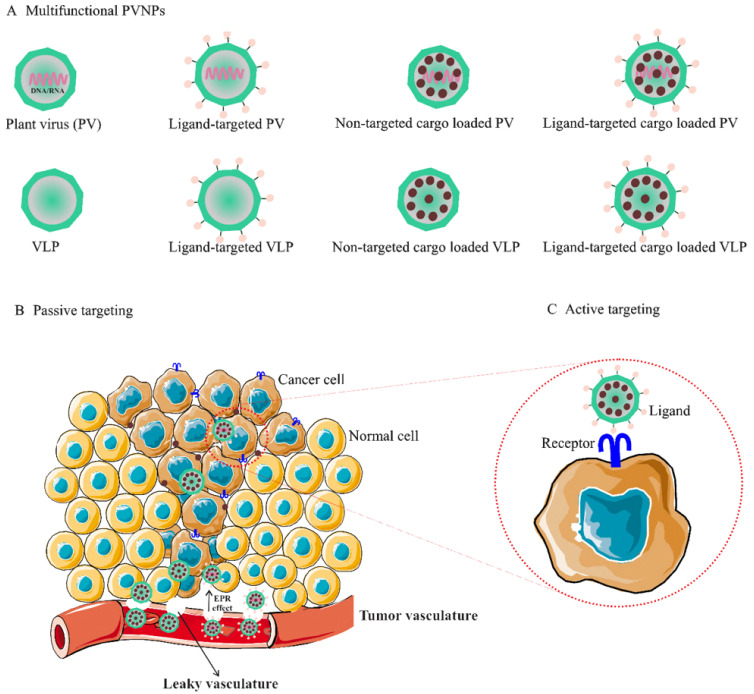
(**A**). Multifunctional PVNPs for breast cancer therapies. PVNPs/VLPs can functionalize by targeting ligands that specifically bind receptors on the surfaces of tumor cells. Non-targeted PVNP/VLP and targeted PVNP/VLP can load anticancer therapeutic agents. (**B**). Non-targeted PVNP/VLP can deliver the therapeutic agents into tumors via the enhanced permeation and retention (EPR) effect (passive targeting mechanism). (**C**). Targeted PVNP/VLP can deliver therapeutic agents via binding to tumor cells’ membrane-bound surface receptors (active targeting mechanism). Abbreviations: PVNPs, plant virus nanoparticles; VLP, virus-like particles.

**Figure 2 vaccines-10-01431-f002:**
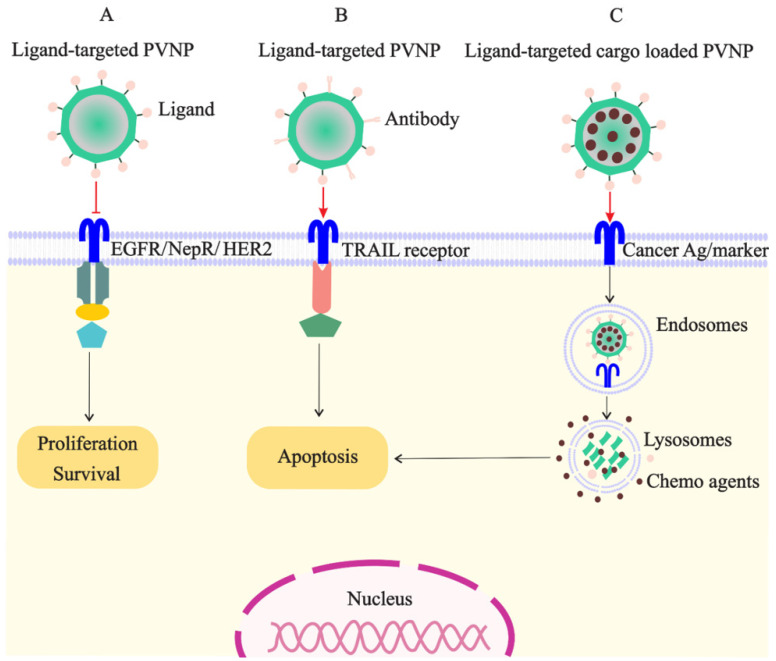
Targeted PVNP/VLP for treatment of breast cancer cells: (**A**). the ligands of receptors overexpressed on cancer cells (e.g., EGFR, NepR, and HER2) can link to PVNPs and inhibit cell survival and proliferation; (**B**). the ligand of TRAIL receptor overexpressed on cancer cells can link to PVNPs to induce apoptosis; (**C**). anticancer agents can load in targeted PVNPs for the internalization and delivery in the cancer cell, which subsequently induces cell death.

**Figure 3 vaccines-10-01431-f003:**
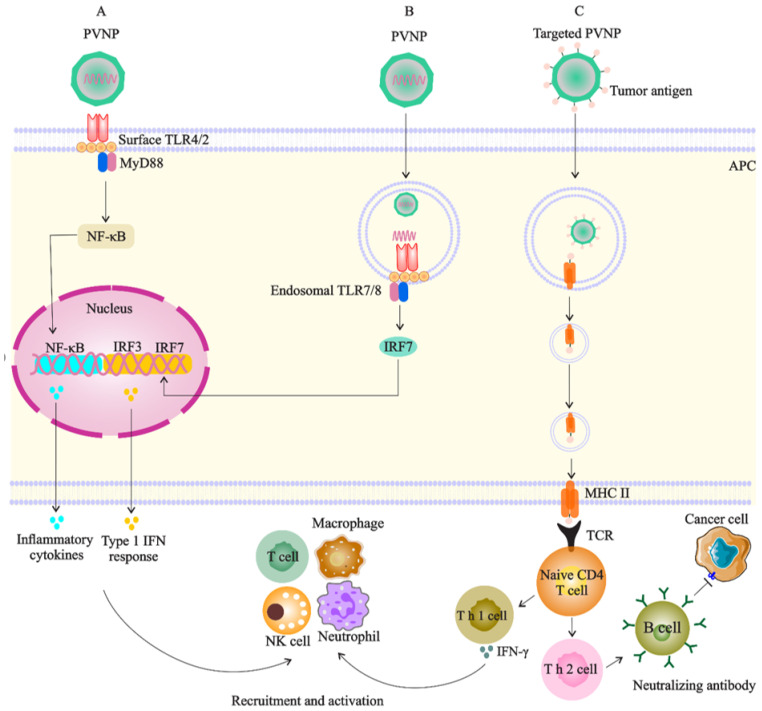
Targeted PVNP/VLP for immune cells (especial DC as APCs): (**A**). PVNP and VNP can bind to surface TLRs of APC and trigger the cytokine secretion to recruit and activate antitumor immune cells; (**B**). the genome of PVNP or VLP loaded with TRL agonists can bind to endosomal TLRs and induce interferon secretion; (**C**). cancer antigens display on the PVNP surface, and phagocytosis by APCs can induce anticancer immune reactions by activation of T cells or lead to the production of neutralizing antibodies against cancer antigens.

## Data Availability

Not applicable.

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
