# Peer review of "Multifunctional Plant Virus Nanoparticles for Targeting Breast Cancer Tumors"

_vaccines, 2022, doi:10.3390/vaccines10091431_

Round 1

Reviewer 1 Report

This is a novel method for delivery. However currently reads as a list, but would benefit from fleshing out more details/examples about how the PVNPs have been used. For example, the TRAIL PVNPs.  All that is stated is that they exist, was it seen in vivo or in vitro? Most of this seems in vitro, how could you get them in vivo? Can mouse models be used? any evidence of clinical trials.  IF there are clinical trials in progress would include a table of those.   If there are mouse studies, give the evidence.  WHAT immune response was induced (type I or type II, how were those evaluated flow, cytokine, etc?).  Also conclusion needs to be more in depth, where is this field going?  What examples of how products are being developed.

Author Response

Reviewer 1

This is a novel delivery method. However currently reads as a list, but would benefit from fleshing out more details/examples about how the PVNPs have been used. For example, the TRAIL PVNPs.  All that is stated is that they exist, was it seen in vivo or in vitro? Most of this seems in vitro, how could you get them in vivo? Can mouse models be used? any evidence of clinical trials.  IF there are clinical trials in progress would include a table of those.   If there are mouse studies, give the evidence. WHAT immune response was induced (type I or type II, how were those evaluated flow, cytokine, etc?). Also, the conclusion needs to be more in-depth, where is this field going? What examples of how products are being developed.

Response: Thank you very much for your suggestion. We changed the structure of the manuscript according to the opinion of the esteemed reviewer. We tried to improve our writing, and believe this version is more compelling and appealing to our audience. We also added more examples of in vivo and in vitro studies as highlighted throughout the text. Overall, plant viruses are manipulatable molecular carriers that can be used to induce the production of cytokines stimulating the anti-tumor immune responses (Figure 3). Notably, they appear to recruit NK cells, activate DCs and stimulate a sustained immune response, and even created immune memory in most mouse models. While they have been in clinical trials for viral diseases (https://doi.org/10.3390/vaccines9070761), PVNP their application for cancer therapy is more limited to preclinical models. However, since the safety profile is so good and efficacy is also shown, we are optimistic that PVNP-ISV will quickly move into active testing in patients. The ISV reaction that disrupts immunosuppressive TME and activates naïve or effector tumor-specific T cells occurs within a few days of intratumoral injection, so ISV with PVNP or other adjuvants could be done in the 1–2 weeks between pathologic diagnosis and surgery that most patients experience.

Reviewer 2 Report

This manuscript reviews the current treatments of breast cancer based on plant nanoparticles. The review mainly focuses on ex vivo models or examples rather than in vivo, which we find insufficient from immunological point of view. The review is very limited when explaining and discussing VLP-based therapy examples. Furthermore, the references are certainly inadequate for the field of VLPs. When discussing VLP-based therapy please elaborate on the design of the therapy and which kind of conjugation was performed (chemical which type/ fusion techniques) as well as the efficacy of the therapy. Overall, the manuscript touches the different topic briefly and there are inadequate explanations. The review mentions the effect of PVNPs on tumor microenvironment in several occasions but failed to explain and discuss this topic clearly and the exact mechanism of action. 

The manuscript needs major revision and elaboration on important topics such as: 

Major revision:

1.     There is a lot of repetition in the manuscript without elaboration on these aspects such as in lines: 37, 78, 330, 347 and others….please list the properties of PVNP in a table or a paragraph and explain why they are better than other nanoparticles as claimed in the manuscript. The authors need to convince the readers of the importance of PVNPs. 

2.     Is Ref 7 the correct reference for specifically plant nanoparticles?

3.     Overall, there are inadequate references.

4.     In paragraph (Multifunctional PVNPs-based tumor therapies) please elaborate on how the plant-nanoparticles can change the tumor microenvironment. 

5.     Please elaborate and explain (passive targeting vs active targeting) for the readers.

6.     Line 82 please explain the mechanism.

7.     Line 92 please explain why filamentous nanoparticles can enhance tumour homing? I would agree that the retention in the tumour is better when using the filamentous vs spherical. 

8.     Line 97 (small molecule drug delivery): I don’t think that giving an in vitro example in lines 100-108 is supportive. Can you add please more in vivo examples to support your claim?

9.     Lines 118-120 please elaborate on how plant nanoparticles overcome the limitations of anti-cancer therapeutic nucleic acid polymers.

10.  Line 122 many nanoparticles (not only plant ones) can be packaged with TLR ligands. Please explain why plant ones are unique in this regard.

11.  Line 129, many of the authors examples are based on in vitro cell line models, which I found insufficient in the field of tumor immunology, please add in vivo examples.

12.  Line 133 please support your claim with additional Refs.

13.  Line 138 please support your claim with Ref, Herceptin is one of the best available treatments.

14.  Kindly improve Figure 1 as it is very hard to visualize the different VLPs.

15.  Line 153, please provide a Ref.

16.  Line 150, PVNPs-based targeted delivery for BC tumors, the whole paragraph lacks Ref?? 

17.  Lines 189-191 please add Refs to support these sentences.

18.  Lines 193-196 please re-phrase.

19.  The same point as before, most of the examples are ex vivo models. Please add more in vivo examples and elaborate on the efficacy of FR-based targeted PVNP.

20.  Lines 215-218 the sentence is not clear? Can you explain please?

21.  Line 219 please elaborate.

22.  Line 230 please elaborate and explain how?

23.  Line 239 please provide an in vivo model or example.

24.  Line 254 please provide an in vivo model or example.

25.  Line 256 explain why and how please.

26.  Line 259 please give examples of how PVNP can be used as vaccine or adjuvant.

27.  Line 263 please give examples on how PVNP can package TLR ligands? Are these particles stable enough to package ssRNA or CpGs? 

28.  Line 266 please give more examples, one Ref is not enough.

29.  Lines 268-275 please add appropriate Refs.

30.  PVNP-based BC monotherapy: Refs are lacking? Certainly not enough Refs, gives the feeling that the paragraph is a personal opinion specially for In situ vaccination.

31.  Line 295 please elaborate.

32.  Kindly list clinical trials if any.

33.  Line 309 please add more supportive Refs.

34.  Line 313 I think it is a very big statement to conclude with very few examples.

35.  Paragraph PVNPs-based combinational therapies for BC, please elaborate.

36.  Figure 3 doesn’t add value to the manuscript. 

37.  Figure 2 needs to be discussed better. 

Author Response

Reviewer 2

his manuscript reviews the current treatments of breast cancer based on plant nanoparticles. The review mainly focuses on ex vivo models or examples rather than in vivo, which we find insufficient from immunological point of view. The review is very limited when explaining and discussing VLP-based therapy examples. Furthermore, the references are certainly inadequate for the field of VLPs. When discussing VLP-based therapy please elaborate on the design of the therapy and which kind of conjugation was performed (the chemical type/ fusion techniques) as well as the efficacy of the therapy. Overall, the manuscript touches on the different topics briefly and there are inadequate explanations. The review mentions the effect of PVNPs on the tumor microenvironment on several occasions but failed to explain and discuss this topic clearly and the exact mechanism of action. 

The manuscript needs major revision and elaboration on important topics such as: 

Major revision:

  1. There is a lot of repetition in the manuscript without elaboration on these aspects such as in lines: 37, 78, 330, 347, and others. please list the properties of PVNP in a table or a paragraph and explain why they are better than other nanoparticles as claimed in the manuscript. The authors need to convince the readers of the importance of PVNPs.

Responses: We thank the reviewer for this informative comment and have modified the manuscript accordingly. The manuscript has been revised and added further relevant details in some other sections as highlighted.

  1. Is Ref 7 the correct reference for specifically plant nanoparticles?

Response: While there is considerable discussion of plant virus nanoparticles, the review includes other type of viruses that can be useful as nanoparticles. This review provides general information introducing nanoparticle modification before the manuscript focuses on PVNPs concepts. 

  1. Overall, there are inadequate references.

Response: We appreciate the feedback and an additional 20 references have been included.

  1. In paragraph (Multifunctional PVNPs-based tumor therapies) please elaborate on how the plant-nanoparticles can change the tumor microenvironment.

Response: To address this important comment, we point out that PVNPs induce the production of cytokines and chemokines that change cold TME into hot TME which subsequently recruit immune cells to tumor sites. This is now added to the section noted and presented in the following sections with more details.

  1. Please elaborate and explain (passive targeting vs active targeting) for the readers.

Response: To address this comment, the active targeting and passive targeting have been explained at the beginning of sections 2 and 3.1. in the revised manuscript (pages 2 and 3). In addition, we try to explain these concepts in revised figure 1.

  1. Line 82 please explain the mechanism.

Response: This is now explained in the revised manuscript as highlighted (page 3, lines 123-127)

  1. Line 92 please explain why filamentous nanoparticles can enhance tumor homing? I would agree that the retention in the tumor is better when using the filamentous vs spherical.

Response: To address this critical comment, this is explained in the revised manuscript (Page 4, lines 138-141).

  1. Line 97 (small molecule drug delivery): I don’t think that giving an in vitro example in lines 100-108 is supportive. Can you add please more in vivo examples to support your claim?

Response: Thank you very much for your suggestion. More examples are explained in the revised manuscript (Page 4, lines 157-161 and 165-167).

  1. Lines 118-120 please elaborate on how to plant nanoparticles overcome the limitations of anti-cancer therapeutic nucleic acid polymers.

Response: To address this critical comment, indeed PVNPs protect the naked RNA/DNA from nucleases degradation in vivo as is explained in the revised manuscript (Page 5, lines 175-177).

  1. Line 122 many nanoparticles (not only plant ones) can be packaged with TLR ligands. Please explain why plant ones are unique in this regard.

Response: We thank the reviewer for this informative comment. Indeed, besides packaging, plant viruses are potent to stimulate TLR activation by their nature which boosts their effect for induce immune responses.

  1. Line 129, many of the authors’ examples are based on in vitro cell line models, which I found insufficient in the field of tumor immunology, please add in vivo examples

Response: Addressing your suggestion, more examples are explained as highlighted on pages 5 and 6.

  1. Line 133 please support your claim with additional Refs.

Response: Addressing your informative comment, more examples are explained as highlighted on pages 5 lines 182-192.

  1. Line 138 please support your claim with Ref, Herceptin is one of the best available treatments.

Response: This claim has been revised, and also the limitation of administration of free Herceptin therapy was explained (page 5 lines 214-215).

  1. Kindly improve Figure 1 as it is very hard to visualize the different VLPs.

Response: We thank the reviewer for that. We have made an effort to improve all figures.

  1. Line 153, please provide a Ref.

Response: Addressing your suggestion, more refs are added.

  1. Line 150, PVNPs-based targeted delivery for BC tumors, the whole paragraph lacks Ref??

Response: Addressing your suggestion, more refs are added.

  1. Lines 189-191 please add Refs to support these sentences.

Response: Addressing your suggestion, more refs are added.

  1. Lines 193-196 please re-phrase.

Response: Addressing your suggestion, mentioned sentences are revised and rephrased.

  1. The same point as before, most of the examples are ex vivo models. Please add more in vivo examples and elaborate on the efficacy of FR-based targeted PVNP.

Response: Addressing your informative comment, more examples are explained as highlighted on pages 7 lines 276-280.

  1. Lines 215-218 the sentence is not clear? Can you explain, please?

Response: We thank the reviewer for that. We have made an effort to improve all text.

  1. Line 219 please elaborate.

Response: To address this comment, this is explained in the revised manuscript.

  1. Line 230 please elaborate and explain how?

Response: To address this critical comment, this is explained in the revised manuscript as highlighted.

  1. Line 239 please provide an in vivo model or example.

Response: Addressing your informative comment, more examples are explained.

  1. Line 254 please provide an in vivo model for example.

Response: Addressing your informative comment, more examples are explained.

  1. Line 256 explain why and how, please.

Response: Addressing your informative comment, more examples are explained.

  1. Line 259 please give examples of how PVNP can be used as a vaccine or adjuvant.

Response: Addressing your critical comment, more examples are explained on pages 10 and 11.

  1. Line 263 please give examples of how PVNP can package TLR ligands? Are these particles stable enough to package ssRNA or CpGs?

Response: Addressing your critical comment, that mechanism of that is explained (page 10 lines 364-368).

  1. Line 266 please give more examples; one Ref is not enough.

Response: Addressing your informative comment, more examples are explained.

  1. Lines 268-275 please add appropriate Refs.

Response: Addressing your informative comment, more examples are explained.

  1. PVNP-based BC monotherapy: Refs are lacking? Certainly, not enough Refs, gives the feeling that the paragraph is a personal opinion especially for in situ vaccination.

Response: We thank the reviewer for that. We have made an effort to improve this section and remove not-necessary subtitles.

  1. Line 295 please elaborate.

Response: Addressing your informative comment, mentioned calm is elaborated.

  1. Kindly list clinical trials if any.

Response: Addressing your valuable comment, until now PVNP-based cancer therapy more ongoing in the preclinical study, however, for viral diseases more studies are reported (ref:10.3390/vaccines9070761)

  1. Line 309 please add more supportive Refs.

Response: Addressing your suggestion, more refs are added.

  1. Line 313 I think it is a very big statement to conclude with very few examples.

Response: Addressing your suggestion, more examples are added on pages 12 and 13.

  1. Paragraph PVNPs-based combinational therapies for BC, please elaborate.

Response: Addressing your valuable comment, the mentioned section is elaborated as highlighted.

  1. Figure 3 doesn’t add value to the manuscript.

Response: Addressing your suggestion, the non-informative fig. 3 is removed.

  1. Figure 2 needs to be discussed better. 

Response: Addressing your suggestion, fig. 2 is now improved and enriched with more information.

Reviewer 3 Report

This paper talks about the using PVNPs targeting breast cancer tumors. Authors describe the related deliveries, targeting, therapies and vaccines and illustrate things using clear figures. Overall, the paper has a high merit due to the high significance of the topic. The paper is also well-written and very organized, which is ready to be published. Just a few things that should be changed: 

1.     Please cite your own figures in your texts. This will help readers refer to the correct figure at the correct time point of reading. 

2.     Some places lack citations, like line 94-96, line 149-167, 189-192, 248-250, 270-271 and so on. Usually, we cite papers after a solid statement. So please go over the entire manuscript and make sure you make appropriate citations after each solid statement. 

3.     Please also check the ordering of the subtitles. Page 7 is numbered with 2, but page 8 continued with 4. You can also re-order the subtitles since 2 point something point something can be confusing. Please try to simply the formatting by re-organizing and summarizing the content of the paper. 

Author Response

Reviewer 3

This paper talks about using PVNPs to target breast cancer tumors. Authors describe the related deliveries, targeting, therapies, and vaccines and illustrate things using clear figures. Overall, the paper has high merit due to the high significance of the topic. The paper is also well-written and very organized and is ready to be published. Just a few things that should be changed: 

  1. Please cite your figures in your texts. This will help readers refer to the correct figure at the correct time point of reading.

Response: Addressing your suggestion, all figs are now cited in texts according to their information.

  1. Some places lack citations, like lines 94-96, lines 149-167, 189-192, 248-250, 270-271, and so on. Usually, we cite papers after a solid statement. So please go over the entire manuscript and make sure you make appropriate citations after each solid statement.

Response: As the respected reviewer mentioned, more related references have been added and all text of the manuscript was improved.

  1. Please also check the ordering of the subtitles. Page 7 is numbered with 2, but page 8 continued with 4. You can also re-order the subtitles since 2 points something point something can be confusing. Please try to simplify the formatting by re-organizing and summarizing the content of the paper. 

Response: We thank the reviewer for this informative comment. Addressing your suggestion, all subtitles are now re-ordered and revised.

Round 2

Reviewer 2 Report

As mentioned previously, the referencing is unbalanced. There are recent important papers un the topic mainly using CuMV VLPs, which is another plant-virus VLP hat has been extensively used in the field. 

Author Response

Please find a revision to your second set of comments. 
